# Coastal Waveform Retracking for Synthetic Aperture Altimeters Using a Multiple Optimization Parabolic Cylinder Algorithm

**Jincheng Zheng** [1,2], **Xi-Yu Xu** [1,*], **Ying Xu** [3,4] **and Chang Guo** [1,2]

1 The CAS Key Laboratory of Microwave Remote Sensing, National Space Science Center, Chinese Academy of Sciences (CAS), Beijing 100190, China; guochang22@mails.ucas.ac.cn (C.G.)
2 University of Chinese Academy of Sciences, Beijing 100049, China
3 Key Laboratory of Space Ocean Remote Sensing and Application, MNR, Beijing 100049, China; xuying@mail.nsosa.org.cn
4 National Satellite Ocean Application Service, Beijing 100082, China
* Correspondence: xuxiyu@mirslab.cn

**Abstract:** The importance of monitoring sea level in coastal zones becomes more and more obvious in the era of global climate change, because, in coastal zones, although satellite altimetry is an ideal tool in measuring sea level over open ocean, but its accuracy often decreases significantly at coast due to land contamination. Although the accuracy of waveform processing algorithms for synthetic aperture altimeters has been improved in the last decade, the computational speed is still not fast enough to meet the requirements of real-time processing, and the accuracy cannot meet the needs of nearshore areas within 1 km from the coast. To improve the efficiency and accuracy in the coastal zone, this study proposed an innovative waveform retracking scheme for the coastal zone based on a multiple optimization parabolic cylinder algorithm (MOPCA) integrated with machine learning algorithms such as recurrent neural network and Bayesian estimation. The algorithm was validated using 153-pass repeat cycle data from Sentinel-6 over Qianliyan Island and Hong Kong–Wanshan Archipelago. The computational speed of the proposed algorithm was four to five times faster than the current operational synthetic aperture radar (SAR) retracking algorithm, and its accuracy within 0–20 km from the island was comparable to the most popular SAMOSA+ algorithm, better than the official data product provided by Sentinel-6. Especially, the proposed algorithm demonstrates remarkable stability in the sense of proceeding speed. It maintains consistent performance, even when dealing with intricate wave patterns within a proximity of 1 km from the coast. The results showed that the proposed scheme greatly improved the quality of coastal altimetry waveform retracking.

**Keywords:** coastal areas; SAR; Sentinel-6; RNN

## 1. Introduction

The oceans, covering 71% of the Earth's surface, play a crucial role in the global material energy cycle and climate regulation [1]. Coastal areas are among the most economically developed regions in the world, with approximately 40% of the world's population living within 100 km of the coast and over 600 million people living at elevations of 10 m or less above sea level [2]. Since the Industrial Revolution, environmental pollution and carbon emissions have led to global warming and a rise in global sea levels [3], which pose a great threat to human survival. The situation is more severe in the last three decades. Coastal residents will be the first to face this severe challenge, as rising sea levels cause problems such as flooding, storm surge, and coastal erosion [4].

Therefore, it is of utmost importance to monitor changes in coastal sea levels and study their impacts. The traditional method for monitoring sea level changes is tide gauge observation [5], but it has limitations such as high construction maintenance costs, lack of representation by single-point observations, and susceptibility to environmental and human factors, making it impractical for achieving a global-scale observation. In

contrast, satellite altimetry technology can provide cost-effective, global, regular sampling observations of the sea surface with high accuracy and all-weather conditions [6]. It has revolutionized the monitoring of global oceans and coastal areas in recent decades.

Historically, most satellite altimeters have been pulse limited, such as the Topex/Poseidon, Jason series developed by US/French, ERS-1/2 and Envisat launched by ESA, and HY-2 series owned by China, along with traditional waveform processing methods [7–9]. Although these conventional methods perform well in open waters and meet basic usage requirements, the interference from land, rain events, and various scattering signals can cause the traditional algorithms to fail or suffer significant accuracy loss [10–13]. This is especially true for altimetry waveforms near the coast, which often perform worse than open ocean waveforms and require more advanced processing.

The introduction of synthetic aperture radar (SAR) technique is a milestone in the history of radar altimetry. In SAR mode, the pulses are arranged as the so-called "bursts". In early implementations such as Cryosat-2 and Sentinel-3, the bursts, including 64 pulses, are transmitted and received every 11.7 miliseconds (for Cryosat-2) or 12.5 miliseconds (for Sentinel-3) with 18kHz pulse repetition frequency (PRF). The mode allows for a reduction in noise by increasing the average number of echoes reflected from the same surface location, alongside a higher signal-to-noise ratio of the received signal [14]. Its high spatial resolution enables it to perform better than traditional altimeters in near-coastal areas.

However, the above implementation, namely, "close-burst mode", has an intrinsic limitation: it cannot exploit the whole observation period. There is no echo in at least half of the time, and the "pseudo" low-resolution measurements by down-sampling the pluses always have poorer performance than that of the traditional radar altimeters. Therefore, the so-called "open-burst mode" were proposed and implemented on the Poseidon-4 radar altimeter embarked on the Sentinel-6 satellite [15]. As shown in Figure 1, the sequence of the pulses was elaborately designed so that interleaved transmitting and receiving pulses can be realized, and this implementation can achieve a low-resolution mode that is fully backward compatible with historical reference elevation measurements, allowing for complete cross-calibration between the latest data and historical records.

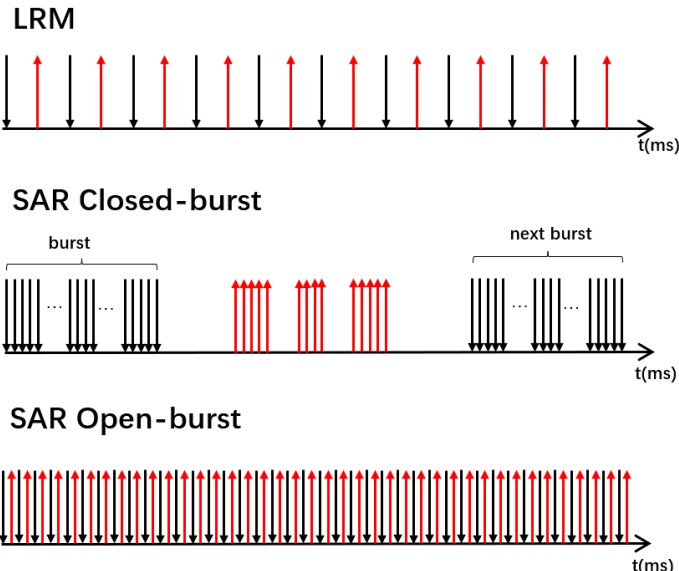

**Figure 1.** Pulse signal transmission methods for the limited pulse radar altimeter (LRM) and synthetic aperture radar altimeter (SAR).

Compared to traditional radar altimeters, synthetic aperture altimeters have further improvements in measurement accuracy, spatial resolution, and application scenarios [16]. Figures 2 and 3 show the echo waveforms of synthetic aperture radar altimeters and traditional pulse-limited radar altimeters within 12 km of the coastline. It can be seen from

the figures that the SAR altimeter waveforms have more obvious statistical characteristics, and waveform contamination only occurs at about 2 km from the coastline. On the other hand, the echo waveforms of the traditional pulse-limited radar altimeter are chaotic and contain many unusable waveforms. Therefore, synthetic aperture radar altimetry has an unprecedented advantage in near-coastal waveform processing.

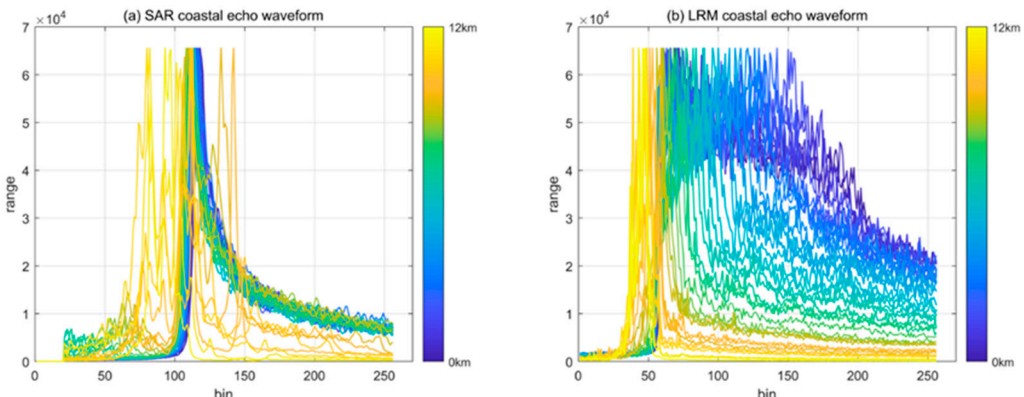

**Figure 2.** Two-dimensional cue diagram of echo waveform 12 km offshore of Sentinel-6: (**a**) SAR and (**b**) LRM.

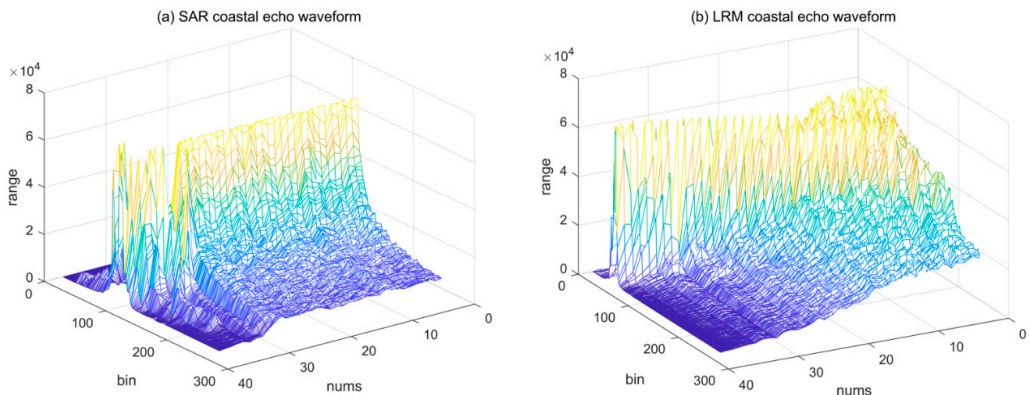

**Figure 3.** Three-dimensional schematic diagram of the echo waveform 0~12 km offshore of Sentinel-6: (**a**) SAR and (**b**) LRM.

Currently, there have been numerous intriguing advancements in the research of synthetic aperture radar altimeter waveform retracking [17–21]. The main algorithms in use include the CryoSat algorithm used in the production of CryoSat satellite level 2 products [22], the MWAPP algorithm proposed by Villadsen et al. [23] for the application of SAR altimetry in inland rivers and lakes, the SAMOSA algorithm proposed by the European Space Agency, and the SAMOSA+ algorithm [24] proposed by Dinardo for improving SAMOSA performance in nearshore areas. While these models provide high accuracy, their complexity and slow computational speed make them unsuitable for (near) real-time applications. Moreover, although some algorithms are designed for coastal and in-land water monitoring, they still have tendency to provide divergent results when the satellite approaches land (within 5 km) [25]. Therefore, in order to realize real-time processing and improve the reliability and feasibility of the algorithm, we adopted a simpler parabolic cylinder model [26] first proposed by Garcia et al. for SAR data processing. The parabolic cylinder model has a simpler structure and faster processing speed, making it more suitable for real-time processing.

However, due to the inherent complexity of the waveform processing algorithms for SAR altimeters, it might be challenging to achieve processing speeds comparable to those of the LRM altimeter's waveform processing algorithm. The processing of echoes

in near-coastal areas is even more challenging, so there are still many problems that need to be solved in this field. Throughout this study, we have designed and evaluated a series of innovative algorithms for SAR altimetry waveform processing. Our investigation begins in Section 2 where we have developed a rapid algorithm that significantly reduces computation time while maintaining excellent accuracy. Building on this foundation, in Section 3, we have tailored the algorithm to nearshore regions by introducing the Modified Parabolic Cylinder Algorithm (MOPCA). In Section 4, we have introduced our selected study area and sources of input data. We have validated the results of our algorithm in Section 5, demonstrating its effectiveness and reliability in accurately processing SAR altimetry waveforms. Finally, in Section 6, we have summarized the key findings of this work and outlined future research directions.

## 2. Parabolic Cylinder Model and Its Accelerated Version Algorithm

The Poseidon-4 radar on board Sentinel-6 has a 9 kHz pulse repetition frequency (PRF), which is approximately four times larger than that of Jason-3, as mentioned in the official Sentinel-6 publication. This high PRF enables Doppler beam sharpening, which generates 64 Doppler beams from a set of 64 echoes [27]. These beams fan out in a sector along the flight direction and allow us to slice the data using the range–azimuth selection from the radar pulse sampling, as shown in Figure 4.

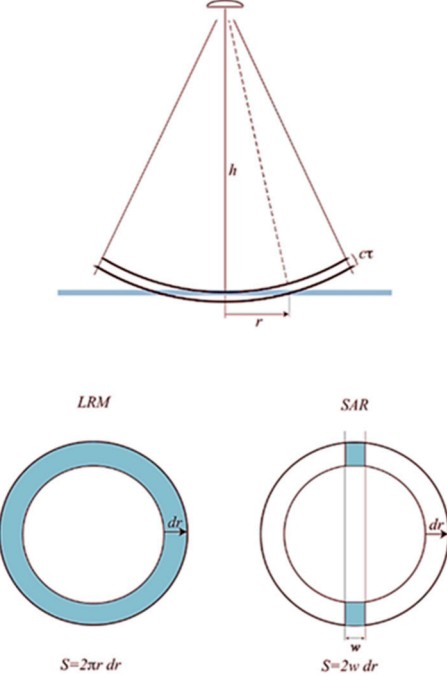

**Figure 4.** Schematic diagram of pulse-limited radar altimeter and synthetic aperture radar altimeter footprints.

In this article, we have used a simpler model that approximates the illuminated sea surface area as the average power of the nadir beam with an effective width, w, along the orbital direction. The illuminated sea surface area can be approximated as follows:

$$S(\tau) \cong 2w \frac{\mathrm{d}r}{\mathrm{d}\tau} H(\tau) \tag{1}$$

where $t_0$ is the arrival time of the waveform, $\tau = t - t_0$, r is the radius of the annulus, and *dr* is the width of the annulus. When $w$ is much smaller than r, it means that the illumination beam pattern can be viewed as close to a rectangle.

$$S(\tau) = w \left( \frac{hc}{\kappa\tau} \right)^{1/2} H(\tau) \tag{2}$$

where *h* is the altitude of the radar antenna above the surface, c is the propagation speed of the radar pulse, $\kappa = 1 + h/R$, and R is the radius of the Earth. The waveform returned by the model is the result of convolving a Gaussian pulse (Gaussian function is a widely adopted approximation of the point target response of the radar system) with the function representing the area under the sea surface versus time.

$$M(\tau) = P(\tau) * G(\tau) * S(\tau) = \frac{wp_0}{\sigma} \sqrt{\frac{2hc}{\kappa\pi}} \int_{-\infty}^{\infty} \exp\left( \frac{-(\tau - \tau')^2}{2\sigma^2} \right) \tau'^{-1/2} H(\tau') d\tau' \tag{3}$$

where $p_0$ is the peak power of a pulse.

By applying an approximation, we obtain the following:

$$M(\tau) = A\sigma^{-1/2} \exp\left( -\frac{\tau^2}{4\sigma^2} \right) D_{-1/2}\left( \frac{-\tau}{\sigma} \right) \exp(-\alpha\tau) \tag{4}$$

Here, $D_a(z)$ refers to the parabolic cylinder function of order a with the argument z. Our objective is to obtain three parameters from the processing: $t_0$, $\sigma$, and $A$.

$$M = A\sigma^{-1/2} \exp\left( -\frac{1}{4}z^2 \right) D_{-1/2}(z) \exp(-\alpha\tau) \tag{5}$$

$$\frac{\partial M}{\partial t_0} = -A\sigma^{-3/2} \exp\left( -\frac{1}{4}z^2 \right) D_{1/2}(z) \tag{6}$$

$$\frac{\partial M}{\partial \sigma} = -A\sigma^{-3/2} \exp\left( -\frac{1}{4}z^2 \right) \left[ \frac{1}{2} D_{-1/2}(z) - z D_{1/2}(z) \right] \tag{7}$$

$$\frac{\partial M}{\partial A} = \frac{M}{A} \tag{8}$$

where $z = -\tau/\sigma$.

The parabolic cylinder model and its derivatives are illustrated in Figure 5:

The parabolic cylinder model was first proposed by Garcia et al. (2014) [26], while it has not been widely adopted in community. The main reason lies in the complexity of the computation of the function $D_a(z)$. It should be noted that $D_a(z)$ in this equation is a hypergeometric function [28]. If the complete analytical solution is used for calculation, the speed would be too slow to meet practical requirements, and the parabolic cylinder model would have no advantage over other models such as SAMOSA.

$$D(a,z) = \frac{1}{\sqrt{\pi}} 2^a / 2 e^{-a^2/4} \left( \begin{array}{l} \cos\left(\frac{\pi a}{2}\right) \varphi\left(\frac{a+1}{2}\right) {}_1F_1\left( -\frac{a}{2}, \frac{1}{2}, \frac{z^2}{2} \right) + \\ \sqrt{2}z \sin\left(\frac{\pi a}{2}\right) \varphi\left(\frac{a}{2} + 1\right) {}_1F_1\left( \frac{1}{2} - \frac{a}{2}, \frac{3}{2}, \frac{z^2}{2} \right) \end{array} \right) \tag{9}$$

where $F_1(a; b; z)$ is a hypergeometric function.

We present a lookup table (LUT)-based algorithm optimization method for fast computation of function $D_a(z)$ values at a = 0.5 or a = −0.5. Two LUTs are created to map the parameter z to its corresponding function value $D_a(z)$, which are then encapsulated into callable functions. Our experimental results demonstrate that this method achieves

comparable accuracy to the original model within a short time span while significantly improving efficiency. Table 1 and Figure 6 display partial information of the LUT.

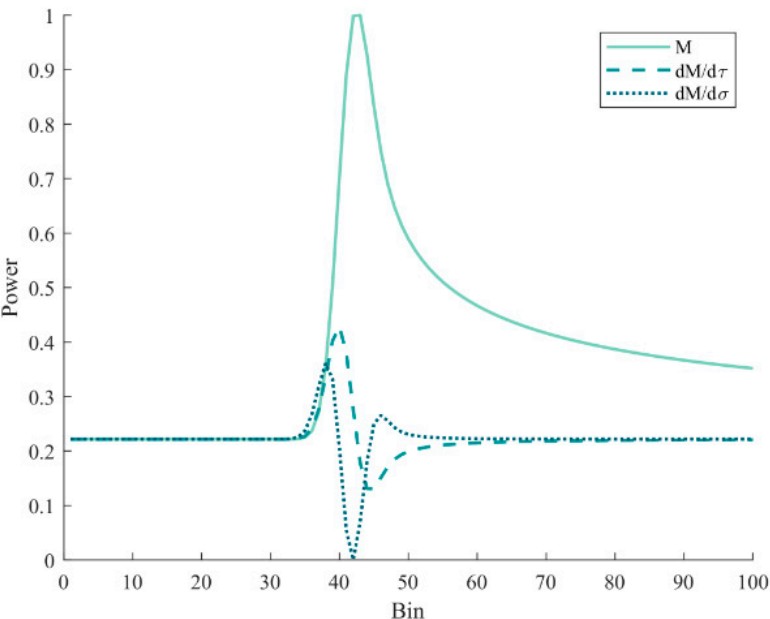

**Figure 5.** Parabolic cylinder model and derivatives (normalized).

**Table 1.** $D_a(z)$ lookup table (excerpt).

| a \ z | 0.1 | 0.2 | 0.3 | 0.4 | 0.5 | 0.6 | 0.7 | 0.8 |
|---|---|---|---|---|---|---|---|---|
| 0.5 | 0.1491 | 0.1581 | 0.1690 | 0.1826 | 0.2 | 0.2238 | 0.2583 | 0.3164 |
| −0.5 | 0.0008 | −0.0010 | −0.0012 | −0.0015 | −0.0020 | 0.0028 | −0.0043 | −0.0079 |

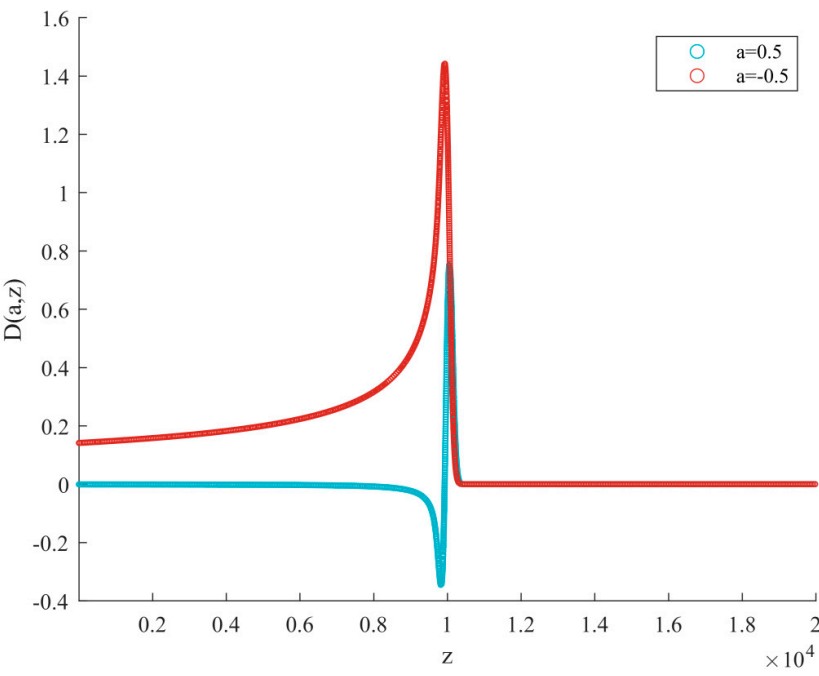

**Figure 6.** $D_a(z)$ lookup table visualization graph with a resolution of $1 \times 10^{-4}$ power. The blue curve is the value of a = 0.5, and the red curve is the value of a = −0.5.

Using this algorithm for fitting, we obtain the results shown in Figure 7.

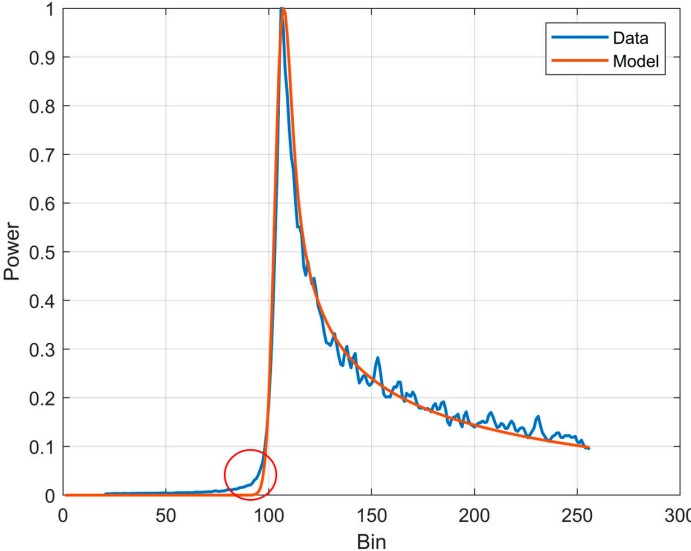

**Figure 7.** Effect diagram of SAR echo wave shape simulated by parabolic cylinder (the blue line represents the original SAR waveform data, and the red line depicts the results after retracking fitting using the parabolic cylinder model).

The fitting results in Figure 7 demonstrate that the proposed algorithm can effectively achieve tracking range. However, Figure 7 also shows a certain degree of offset in the rising edge. We believe that this is not caused by the lookup table optimization algorithm, but by the error introduced by approximating the actual sinc(·) point target response function with a Gaussian function in the parabolic cylinder model. It should be noted that this approximation strategy is not only present in the parabolic cylinder model, but also adopted by many SAR processing models, indicating that it is a common issue [29]. Figure 8 shows the influence of Significant Wave Height (SWH) and off-nadir angle on the shape of the backscattering model. The front edge of the backscattering model is mainly affected by SWH. Therefore, we consider this offset to be closely related to SWH and propose a new algorithm to eliminate it, which will be detailed in Section 4.

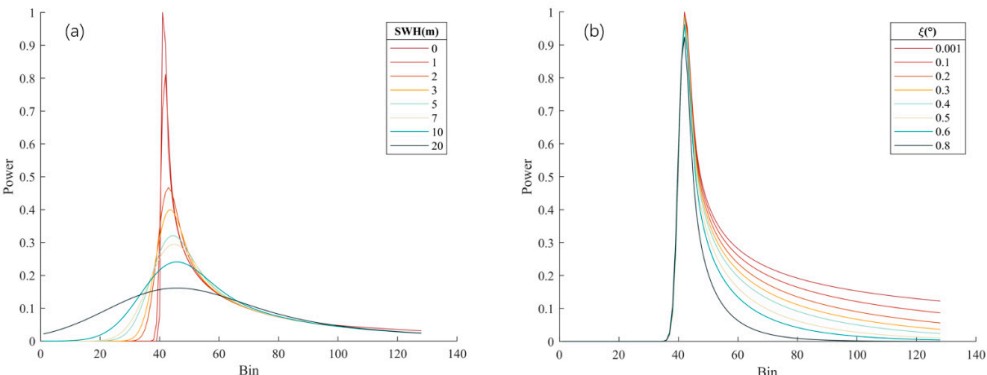

**Figure 8.** The impact of SWH and off-nadir angle on the shape of the backscattering waveform: (**a**) shows the effect of SWH on the backscattering waveform and (**b**) shows the effect of off-nadir angle on the backscattering waveform.

During the testing phase, we conducted experiments using simulated waveforms. We generated ten thousand simulated backscattering waveforms and used five different algorithms for retracking processing. These algorithms included MLE3 [30], SAMOSA-semi-analytical model, SAMOSA2 model [31], parabolic cylinder-analytical form, and parabolic cylinder-LUTs. Under the same software and hardware platform conditions with

a CPU of i5-12600 (Intel Semiconductor, Santa Clara, CA, USA), we recorded the running time of each algorithm and presented the results in Table 2.

**Table 2.** Time consumption statistics for five algorithms in retracking range.

| Analog Waveform Form | Algorithm | Number of Waveforms Successfully Processed | Total Time(s) | Average Time (s) |
|---|---|---|---|---|
| LRM | MLE3 | 8392 | 89 | 0.00917 |
| SAR | SAMOSA-semi-analytical model | 9541 | 1032 | 0.1082 |
| SAR | SAMOSA2 model | 9532 | 583 | 0.0612 |
| SAR | Parabolic cylinder-analytical form | 9607 | 420 | 0.0437 |
| SAR | Parabolic cylinder-LUTs | 9607 | 118 | 0.0111 |

To test the stability (algorithms in the sense of different waveform processing speed) of the algorithms in handling different waveforms, we divided 10,000 simulated waveforms into five groups (2000 waveforms each), which are independent of each other and have different levels of quality. In the experiment, we recorded the time required for five algorithms to process each group (2000 waveforms) separately. We obtained stability curves for various algorithms by recording the ratio of the time required for each algorithm to compute every two thousand waveforms to the total time. The Figure 9 show that the two parabolic cylinder algorithms perform more stable than other algorithms, with little fluctuation even when processing complex waveforms. This indicates that the parabolic cylinder algorithms are better suited for real-time processing requirements.

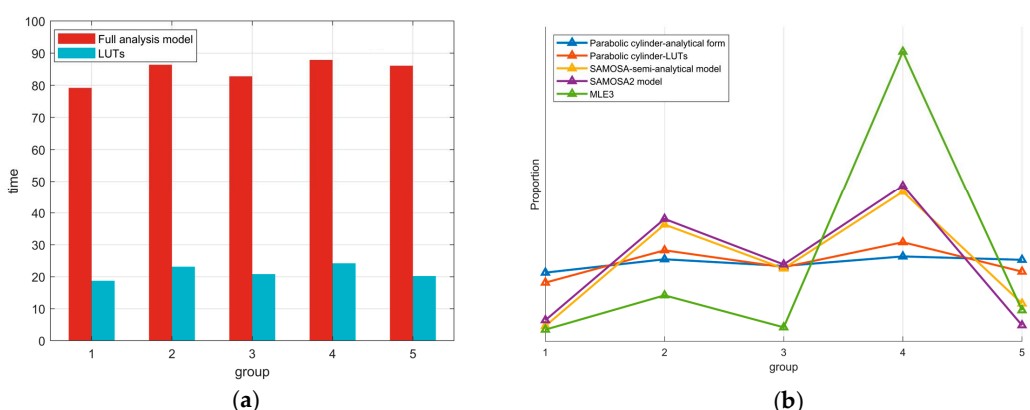

(**a**)                                                                                    (**b**)

**Figure 9.** Results of grouped tests (groups one to five, each had two thousand separate test waveforms): (**a**) time required by the two parabolic cylinder algorithms for every 2000 waveforms and (**b**) proportional time required by each algorithm for processing 2000 waveforms out of the total.

## 3. Coastal Retracker Strategy Based on the Parabolic Cylinder Model

### 3.1. Integrated Recurrent Neural Network Algorithm

At present, both SAR and LRM waveform retracking have good performance in open sea. However, due to the specific characteristics of coastal areas, the use of basic algorithms often results in significant errors or even fail to converge. To address this issue, we propose a new algorithm, the multiple optimization parabolic cylinder algorithm (MOPCA), specifically designed for nearshore areas. Our improved version has four major improvements over the conventional parabolic cylinder algorithm: the introduction of the recurrent neural network (RNN) model, a two-step retracking calculation, Bayesian parameter estimation, and RNN-based error correlation. Figure 10 shows the flowchart of the entire algorithm.

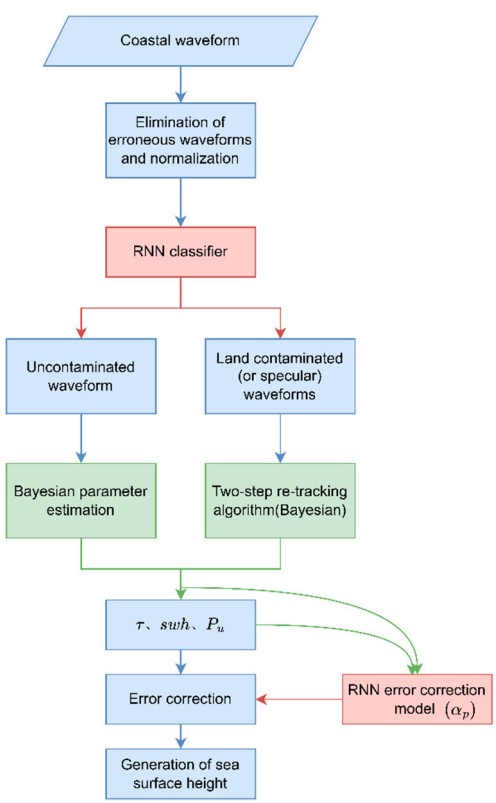

**Figure 10.** Flowchart of the nearshore parabolic cylinder algorithm.

### 3.1.1. Waveform Classification

Coastal radar waveforms have diverse forms compared to the waveforms observed in open ocean, mainly due to the influence of geographic features such as marine pollutants, river mouths, islands, and complex nearshore terrain. These factors cause complex waveforms with varying characteristics, which make it a challenging task to classify coastal waveforms for effective monitoring and assessment of marine environments and changes. This has been an ongoing focus of remote sensing research. To address this issue, we used a graphic processing approach to analyze coastal radar echoes and classified them into two categories: standard SAR echoes and peak echoes contaminated by land obstructions. We employed a novel classification strategy, the RNN algorithm, for these two waveform classifications. The RNN algorithm showed better accuracy and more robust performance than commonly used classification methods based on waveform features.

RNN is a type of neural network that can process input sequences and output sequences with advantages such as memorization, parameter sharing, Turing completeness, and more. It can extract temporal and semantic information that is useful for learning and classifying complex sequence data [32], especially when it comes to processing time series data [33]. RNN algorithm performed exceptionally well when processing oceanic environmental data with temporal and cyclical patterns. Therefore, we used the RNN algorithm to classify coastal radar waveforms and effectively improved the accuracy of classification. The RNN algorithm can handle time series information in marine environmental data and is expected to play a significant role in waveform retracking, providing technical support for monitoring and studying coastal environmental changes.

Figure 11 shows a schematic diagram of the recurrent neural network (RNN), which explains how the hidden layer at time t is influenced by both the input layer and the hidden layer at time t−1. This mechanism makes RNN more sensitive to sequential data and appropriate for handling time series data. RNN has various structures, such as one-to-many, many-to-many, and Encoder–Decoder [34], which have their own specific applications. The appropriate structure should be selected for specific problems. For

example, the one-to-many structure is suitable for generating sequential output from a single input, such as image captioning [35]. The many-to-many structure is suitable for scenarios where both input and output are sequences, such as video classification [36]. The Encoder–Decoder structure is suitable for cases where the lengths of input and output sequences are different, such as machine translation [37].

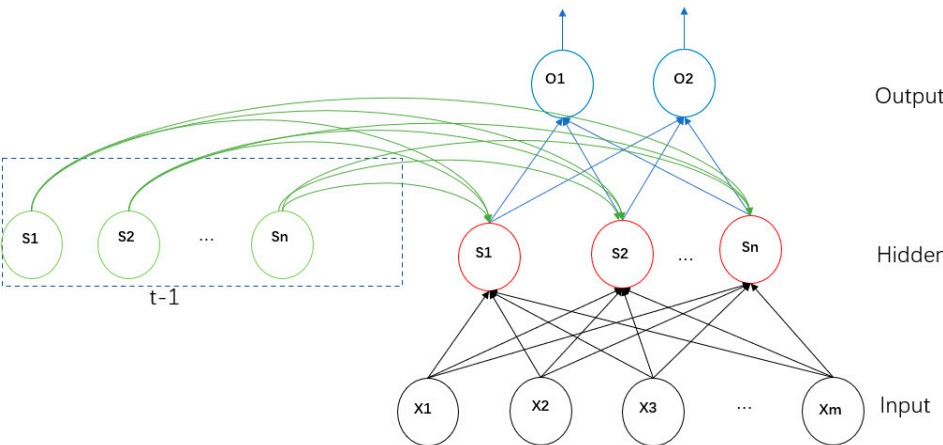

**Figure 11.** A schematic diagram of a simple RNN algorithm.

For our waveform classification problem, we eliminated the unusable noise waveform (5%) in the waveform and divided them into two categories: oceanic return echoes and peak echoes contaminated by land noise. As shown in Figure 12,this resulted in two possible classification results, 0 and 1. We chose the many-to-many structure based on the RNN for waveform classification, which is more appropriate for our problem. Each input waveform corresponds to a classification label, and a classification result vector is attained. This method not only improves classification accuracy but also has more potential for future remote sensing research and applications in the marine environment.

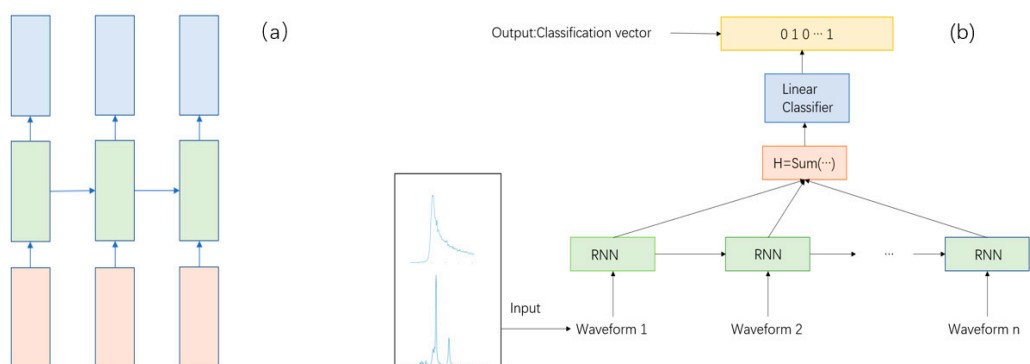

**Figure 12.** Figure depicting RNN classification algorithms: (**a**) conceptual diagram of the n-to-n RNN model structure and (**b**) schematic diagram of an RNN algorithm designed for echo classification.

We divided 10,000 waveforms into training and testing sets at a ratio of 9:1 and conducted classification experiments, and the results are shown in Figure 13. After 30 iterations, the loss reached stability, and the final classification accuracy was nearly 98%, both in the training and testing sets. These results indicate that the proposed RNN-based many-to-many structure algorithm performs excellently in classifying waveforms. Furthermore, by comparing the training and testing sets, we can conclude that the model has a good generalization ability for new samples.

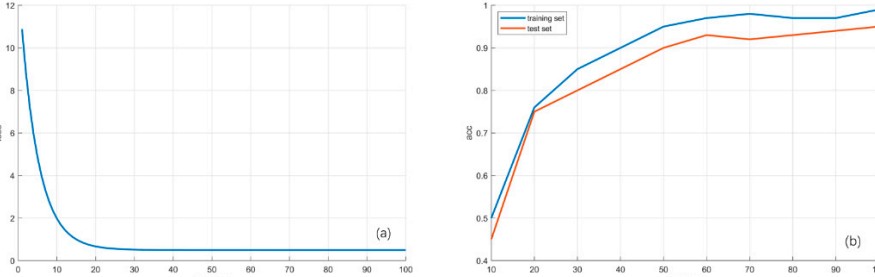

**Figure 13.** Schematic diagram of RNN training results: (**a**) graph showing the change in loss function with iteration and (**b**) graph showing the change in accuracy of the training and test sets with iteration.

### 3.1.2. Computational Model Based on RNN

The second application of the recurrent neural network algorithm in our processing of nearshore echo waveforms is to eliminate the errors caused by using a Gaussian function instead of the sinc function. Due to the complexity and importance of the problem, it is necessary to write an individual article to clarify this issue and present the details, and this paper presents only the solution approach. Detailed discussions on algorithmic details are under preparation and will be provided in another article of the authors.

$$\text{sinc}^2(x) \approx e^{-\left(\frac{x}{\sqrt{2}\alpha_p}\right)^2} \tag{10}$$

The current approximation method leads to significant errors in the calculated sea surface height. To mitigate these errors, it has been proposed to use different $\alpha_p$ parameters for different sea conditions. The commonly used method is to create a lookup table of $\alpha_p$ parameters and SWHs, which are used to eliminate the error term during the final calculation of the sea surface height. However, this method faces certain problems. Firstly, a small step size is required, which necessitates a large physical space as the data increases. Secondly, this method cannot obtain continuous $\alpha_p$ parameter, and interpolation may also be required, compromising the original goal of automated processing. Therefore, we propose to use a lightweight model trained using a recurrent neural network (RNN) to solve for the $\alpha_p$ parameter. The input parameters of the RNN model include the 128-dimensional vector of echo waveforms and SWH, and the output parameter is the beta value. This approach is a novel attempt to solve this type of problem using neural networks. The algorithm can provide more accurate and continuous results.

### 3.2. Two-Step Retracking

The closer to the shoreline, the more and pollution (or specular reflection) the waveform involves. For severely polluted cone-shaped waveforms, the use of basic retracking algorithms would lead to significant loss of lock. In order to address this issue, we have developed a two-step retracking algorithm that takes into account the classification results of a RNN in the first step. If the waveform is not polluted or has only minor pollution, we still use the parabolic cylindrical algorithm. For waveforms affected by land pollution (or mirror reflection), we adopt the two-step retracking algorithm, during which the waveform is first processed using the parabolic cylindrical algorithm in the first step. However, in the second step, we set the SWH to 0 [38], and again use the parabolic cylindrical algorithm to calculate the sea surface height.

The fitting results of the two-step retracking algorithm are shown in Figure 14. Due to the consistent use of the parabolic cylindrical model, the resulting sea surface height is virtually identical for both ordinary ocean echo waveforms and peak waveforms. This step is crucial for generating continuous nearshore sea surface height data.

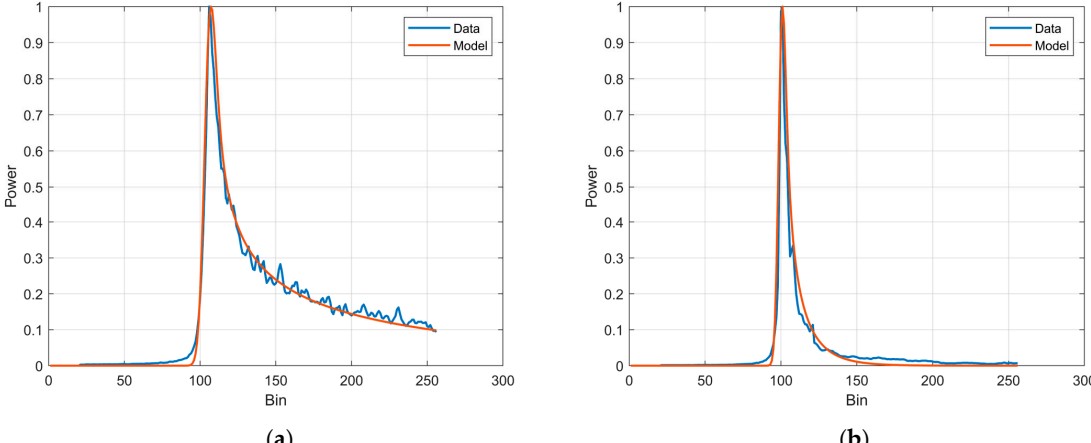

(**a**)                                     (**b**)

**Figure 14.** The fitting results of waveform data: (**a**) shows the fitting using single retracking method for unpolluted echo waveform and (**b**) shows the fitting using two-step retracking method for severely polluted cone-shaped waveform.

### 3.3. Bayesian Parameter Estimation

In the final and most important step, we abandoned the traditional least-squares algorithm in favor of Bayesian estimation for echo parameter estimation in Algorithm 1. It is well known that the calculation complexity of the least-squares algorithm is low, but it is sensitive to data with high noise [39], which can lead to overfitting. Nearshore data often contains high levels of noise, which can result in relatively large errors when using the least-squares algorithm and a corresponding loss of accuracy. As a parameter estimation algorithm, Bayesian estimation can combine prior probabilities with likelihood functions to estimate posterior probability distributions of the parameters when they are unknown [30]. It can mitigate the problems of noisy data and overfitting and can better utilize sample information to obtain more reliable model parameters.

However, the current popular SAR echo models are very complex, making it difficult to obtain a mathematical description of the Bayesian formula, which can lead to a slow solving process that does not meet practical requirements. Fortunately, the parabolic cylindrical model we use perfectly solves this problem. The expression for the parabolic cylindrical model is very concise, and the loss of time efficiency from using Bayesian estimation can be completely ignored. However, it can greatly improve the poor retracking accuracy of coastal data.

---

**Algorithm 1** Bayesian parameter estimation

---

1. Initialize the parameters $\theta = \{\theta_1, \theta_2, \theta_3\}$
2. Set the number of iteration times T, and initialize the parameter sample set S
3. For t = 1 to T:
3.1 Calculate the likelihood function $P(D|\theta)$ and the posterior probabilit $P(\theta|D)$ based on the current parameter $\theta$
3.2 Sample a new parameter $\theta'$ from the posterior distribution $P(\theta|D)$
3.3 Add $\theta'$ to the parameter sample set S.
3.4 Update the current parameter $\theta$ to $\theta'$ and continue the loop.
4. Calculate the estimated values of the parameters:
4.1 For each parameter $\theta_i$, calculate its sample mean $\bar{\theta}_i$ as the estimated value.
5. Output the estimated values of the parameters $\bar{\theta}_1, \bar{\theta}_2$ and $\bar{\theta}_3$

---

## 4. Study Areas and Data

We selected two areas, Hong Kong (HK)–Wanshan Archipelago and Qianliyan Island, to study the accuracy and reliability of Sentinel-6 satellite altimeter data in coastal zone.

As shown in Figure 15, the 153 pass of the satellite covers both areas, allowing for a synchronous comparative analysis.

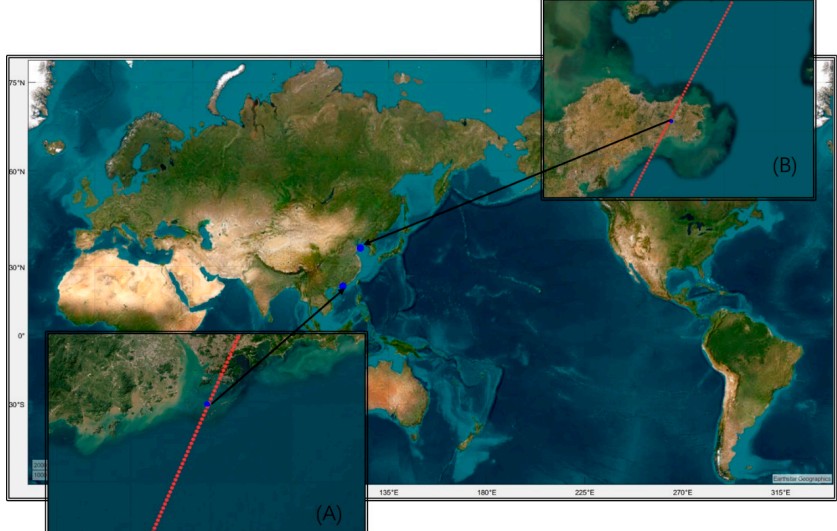

**Figure 15.** Schematic diagram of Sentinel-6 153 pass crossing the HK–Wanshan Archipelago (area (**A**)) and Qianliyan Island (area (**B**)).

The Wanshan Islands near HK are located just south of the Tropic of Cancer, and have extremely complex geographical features, consisting of numerous islands. The climate is predominantly subtropical and displays clear seasonal variations [40]. On the other hand, Qianliyan Island is a small island located on the continental shelf on the western coast of the central part of the Yellow Sea, and the closest distance to the mainland is about 45 km, which indicates a better coastal altimetry performance. The area has undergone changes over time and was once referred to as the collective name for over 100 islands in the open sea. Based on the Sentinel-6 altimeter orbit and the matching of the tide gauge measurements, we used the HK–Wanshan offshore (area A) in the range of 21.5°N–22.5°N and the Qianliyan offshore (area B) in the range of 36°N–38°N as the study areas.

To fulfill the requirements waveform classification and retracking, we used the L1B and L2 level data of 35 cycles from Sentinel-6 between 2022 and 2023 as our data source. Both HR (high-resolution) and LR (low-resolution) data were processed. The detailed SAR mode waveforms for the two areas are shown in Figure 16, with increasing numbers indicating waveforms closer to the coastline. Apart from a few noisy waveforms (green), they can be mainly classified into two types: ocean-like waveforms (red) and single-peak waveforms affected by land (blue).

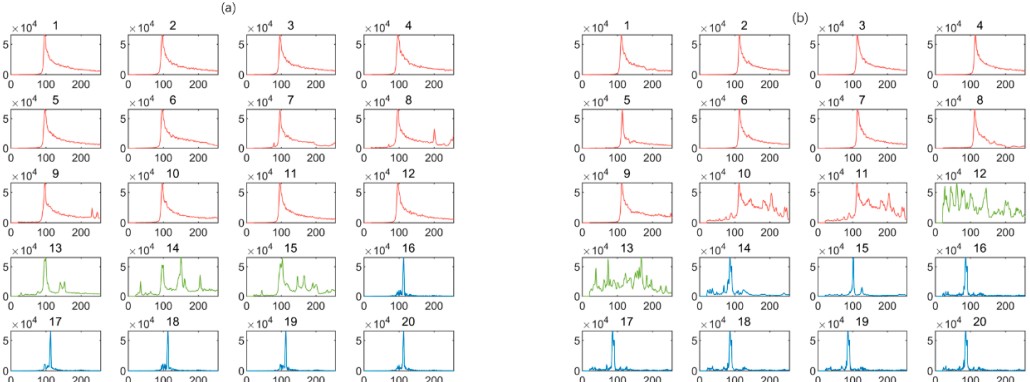

**Figure 16.** Schematic diagrams of sample echoes in the study area: (**a**) waveform schematic of 153 pass crossing the HK–Wanshan Archipelago and (**b**) waveform schematic of 153 pass crossing Qianliyan Island.

As described by a previous study [41], the Quarry Bay tide gauge in HK is chosen for the validation of the algorithms. It provides sea level data with an accuracy of 1 cm for a single measurement and is regularly calibrated every other year. The tide gauge is located at 114.22°E, 22.28°N, near the northern coast of the HK Island, separated from the Kowloon Peninsula by the Victoria Harbor. Sea level on this area is likely influenced by anthropogenic local-scale factors, in addition to more regional and global ocean variations. Hourly tide gauge data were used in this study.

## 5. Results

This section shows the validation results of MOPCA using the repetitive cycle data of Sentinel-6. The algorithm is compared with current representative approaches including MLE3, ALES+, CryoSat, MWAPP, and SAMOSA+. It is important to note that MLE3 and ALES+ algorithms are designed specifically for pulse-limited altimetry data. For these algorithms, the waveforms in low-resolution mode are processed.

### 5.1. Efficiency Analysis

To evaluate the performance of the algorithms in offshore areas, we classified 35 cycles of repetitive data according to offshore distance. We then processed the data using the six waveform tracking algorithms and recorded the processing time on the same computing platform (i5-12600 CPU, Intel Semiconductor, Santa Clara, CA, USA). This allowed us to analyze the efficiency and accuracy of the algorithms in processing echo data in offshore areas.

As shown in Figure 17, the processing time required for waveform tracking of the six algorithms varies with offshore distance. It is evident that current representative SAR waveform processing algorithms require significantly more time than LRM waveform processing algorithms. However, our proposed MOPCA algorithm boasts an efficiency that is comparable to representative LRM waveform processing algorithms. Notably, it even outperforms the ALES+ algorithm in the more complex offshore areas. This highlights the potential of our MOPCA algorithm for improving SAR waveform processing in near-real-time remote sensing applications.

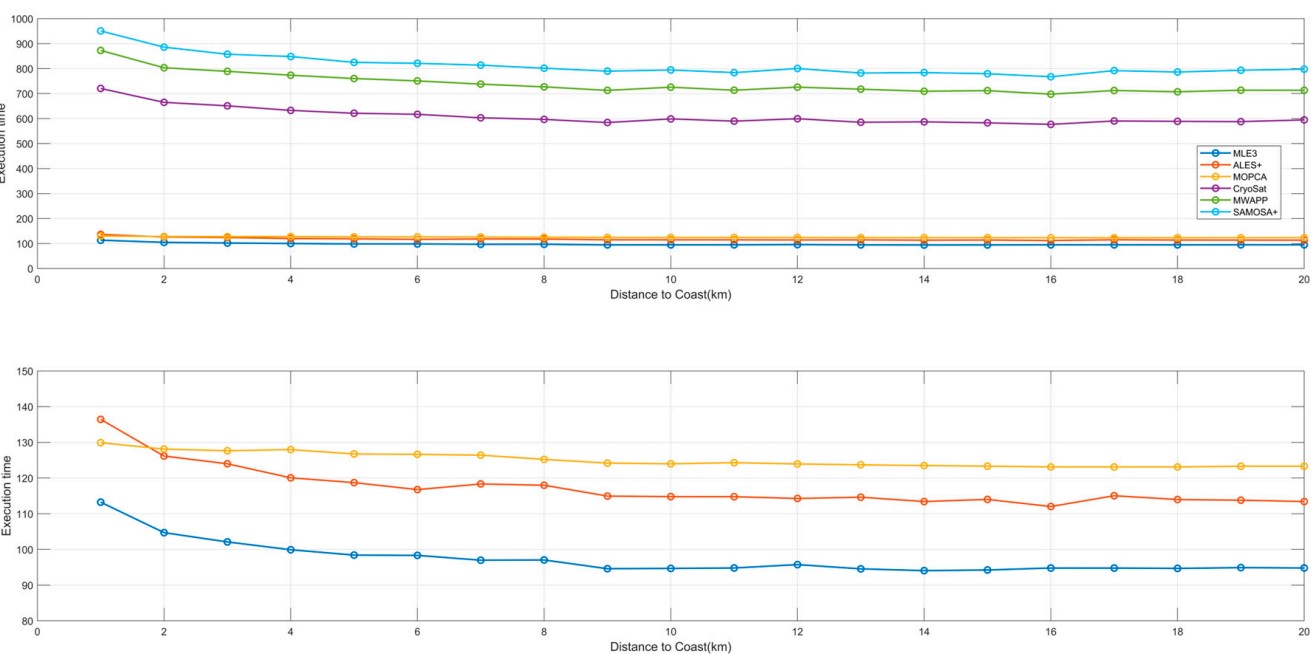

**Figure 17.** A processing time graph depicting the variation in six different algorithms with respect to the offshore distance.

As shown in Figure 18, the processing time of the algorithms is plotted in groups of every five kilometers. It is evident that there is a significant difference in processing time as offshore distance varies, regardless of whether the algorithm is designed for LRM or SAR waveform echo processing. Our proposed MOPCA algorithm, however, remains stable at around 12 s per 1000 waveforms, highlighting its strong stability and consistent performance.

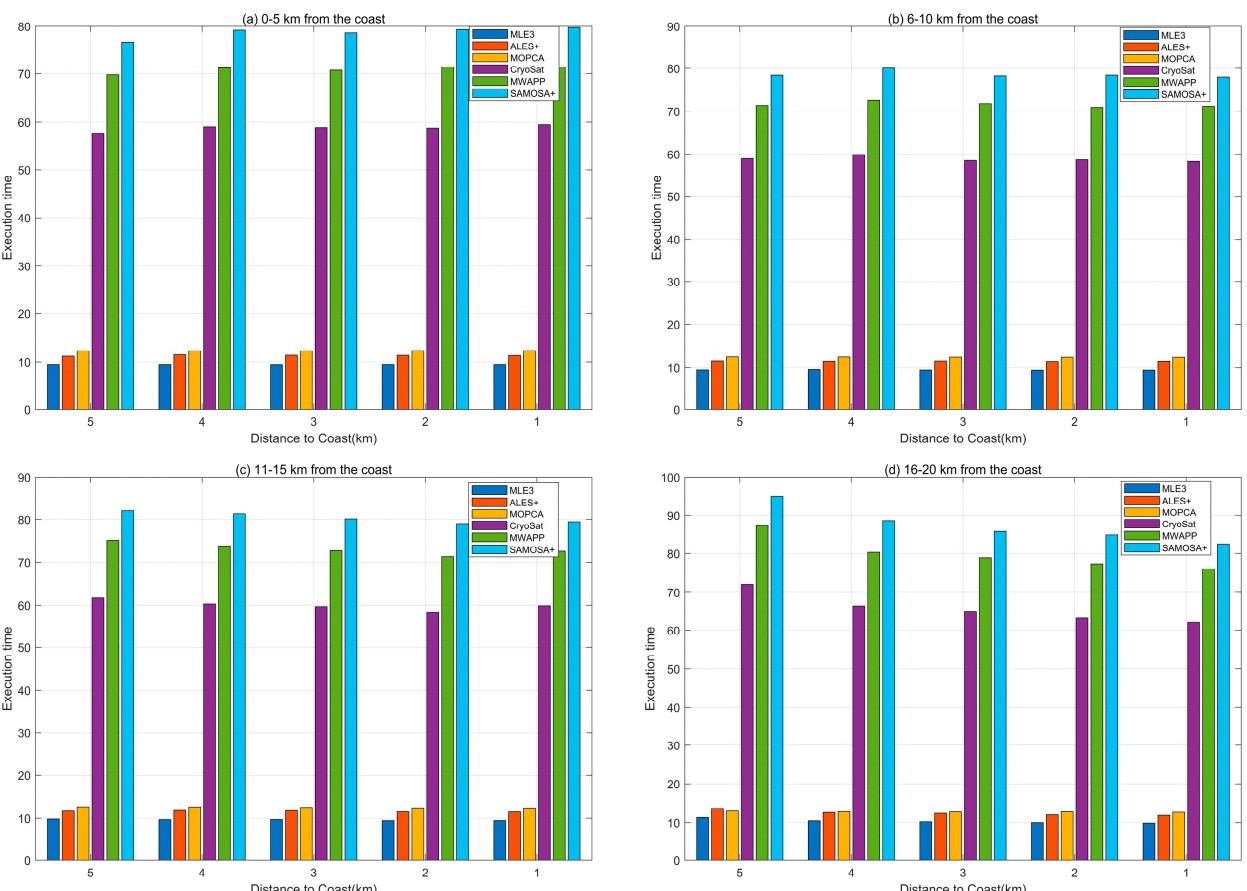

**Figure 18.** A comparison chart of echo processing times for six algorithms with an interval of five kilometers.

### 5.2. Precision Analysis

In this study, the accuracy evaluation of the algorithm uses the following three indicators: (1) Retracking success rate. When the coastal waveforms are contaminated by land, the waveform retracking algorithm may fail to extract the sea surface height information or produce biased results. We can evaluate the retracking success rate by calculating the ratio of the successfully retracked waveforms to the total waveforms. (2) Standard deviation of the sea surface height of the retracked altimetry results. The waveform retracking in coastal areas is more affected by land contamination than in open ocean areas, resulting in higher standard deviation, which reflects the instability of the sea surface height. Therefore, we can use the mean standard deviation of the retracked results to measure the performance of different algorithms in processing contaminated data. The smaller the standard deviation, the better the retracking performance. (3) Correlation and RMSE difference with tide gauge data. To verify the external reliability of the retracking algorithm, we can calculate the mean correlation and RMSE difference between the time series of retracked sea surface height and tide gauge sea surface height at every 1 km location, and only use the sea surface height data with the same observation time for comparison.

5.2.1. Retracking Success Rate Analysis

We applied a three-sigma outlier rejection to the 35 retracked sea surface height time series from six algorithms in the coastal areas of HK–Wanshan Archipelago and Qianliyan Island within 1–20 km from the shore and obtained the retracking success rates of different algorithms at different distances from the shore, as shown in Table 3. From the table, we can see that the retracking algorithms based on LRM waveforms have low success rates in the coastal areas, while the retracking algorithms for SAR waveforms perform better. Especially, our proposed MOPCA algorithm shows high performance in both Qianliyan and HK–Wanshan areas. Although this algorithm has similar performance to the advanced SAMOSA+ algorithm at distances above 10 km from the shore, it outperforms SAMOSA+ algorithm within 10 km from the shore, demonstrating its advantage in nearshore processing.

**Table 3.** Retracking success percentage.

| Area | Distance to Coast/km | Retrackers | | | | | |
|---|---|---|---|---|---|---|---|
| | | MLE3 | ALES+ | MOPCA | CryoSat | MWAPP | SAMOSA+ |
| HK–Wanshan Archipelago | 0~1 | 34.54% | 41.02% | 62.00% | 50.41% | 52.29% | 59.01% |
| | 1~2 | 42.22% | 52.47% | 79.32% | 70.47% | 74.59% | 73.15% |
| | 2~4 | 82.46% | 85.12% | 92.18% | 89.70% | 88.31% | 90.20% |
| | 4~6 | 92.54% | 93.58% | 96.78% | 94.23% | 93.93% | 95.52% |
| | 6~8 | 94.03% | 95.03% | 97.89% | 95.44% | 95.62% | 97.31% |
| | 8~10 | 94.53% | 96.38% | 98.71% | 96.29% | 96.01% | 98.34% |
| | 10~15 | 95.47% | 97.23% | 98.93% | 97.16% | 97.74% | 99.03% |
| | 15~20 | 96.57% | 98.65% | 99.17% | 99.35% | 98.97% | 99.55% |
| Qianliyan Island | 0~1 | 38.26% | 43.21% | 64.20% | 51.26% | 53.03% | 59.25% |
| | 1~2 | 44.19% | 56.10% | 80.16% | 72.19% | 74.93% | 75.28% |
| | 2~4 | 84.07% | 86.33% | 91.90% | 89.11% | 89.24% | 90.98% |
| | 4~6 | 93.54% | 92.89% | 96.42% | 95.04% | 94.21% | 95.77% |
| | 6~8 | 94.61% | 96.04% | 98.28% | 96.05% | 95.97% | 97.09% |
| | 8~10 | 94.52% | 96.31% | 98.94% | 96.64% | 97.01% | 98.64% |
| | 10~15 | 95.27% | 97.19% | 98.91% | 97.23% | 97.54% | 99.18% |
| | 15~20 | 97.37% | 98.23% | 99.35% | 98.72% | 98.15% | 99.57% |

5.2.2. Standard Deviation of the Sea Surface Height of the Retracked Altimetry Results

We computed the difference between the sea surface height calculated by six algorithms and their mean elevation as a function of distance from the shore. The results show that except for the two algorithms for LRM mode, the SAR waveform processing algorithms perform well, especially the MOPCA algorithm, which uses an optimized algorithm for nearshore areas, and does not have much fluctuation even within 5 km from the shore. To ensure enough along-track sampling, we combine the data from both study areas and calculate the standard deviation at 5 km intervals. The results are shown in Table 4. According to the results in the table, we can see that for the SAR mode retracking algorithms, except for the 10–15 km interval where the MOPCA algorithm has slightly worse standard deviation than the SAMOSA+ algorithm, and the MOPCA algorithm performs best in the other intervals. From the overall standard deviation situation, the MOPCA algorithm is more suitable for processing nearshore waveforms.

**Table 4.** Standard deviation of algorithms calculated in groups of 5 km (cm).

| Distance to Coast (km) | MLE3 (LRM) | ALES+ (LRM) | MOPCA (SAR) | CryoSat (SAR) | MWAPP (SAR) | SAMOSA+ (SAR) |
|---|---|---|---|---|---|---|
| 0–5 | 12.24 | 8.34 | 3.13 | 7.31 | 8.78 | 6.63 |
| 5–10 | 9.51 | 7.0 | 2.52 | 5.96 | 4.27 | 4.72 |
| 10–15 | 8.29 | 6.63 | 2.03 | 3.67 | 3.91 | 1.73 |
| 15–20 | 4.59 | 4.92 | 1.87 | 3.06 | 3.62 | 1.67 |

### 5.2.3. Correlation with Tide Gauge Data

The altimeter data utilized in this study were derived from Sentinel-6's L1B dataset, as detailed in Section 4. Our geophysical corrections encompass dry and wet tropospheric components, ionosphere correction, sea state bias correction, solid earth tide correction, polar tide correction, ocean tide correction, and load tide correction and dynamic atmospheric correction [42]. These correction parameters are extracted from the official dataset. We rigorously selected observation data from tide stations located within 20 km of the Sentinel-6 satellite ground track observation point, ensuring high data quality. This effort resulted in the compilation of 18 sets of data at ten-day intervals spanning from January 2022 to June 2022. Additionally, to ensure data coherence, dynamic atmospheric corrections were applied. Employing retracking algorithms, the nearshore observation data were recalibrated to derive accurate sea surface height information. Simultaneously, the EGM2008 model was employed to rectify geoid gradient variations at each point relative to the nearest tidal survey station. The obtained findings were categorized based on offshore distance: 0–5 km, 5–10 km, and 10–20 km. Ultimately, by quantifying the differences between sea surface height time series at individual points and corresponding values from the nearest tide station, we validated the accuracy of the distinct algorithms employed in this study.

From Figure 19, it is evident that, within the range of 10–20 km from the coast, most of the algorithms exhibit high consistency with the observed values at the tidal gauge station, except for occasional jumps in the results obtained through traditional methods. However, within the range of 5–10 km, the performance of traditional LRM algorithms is insufficient to meet the requirements, especially with the MLE3 algorithm resulting in significant fluctuations. At the closest range of 5 km from the coast, the performance of all algorithms is suboptimal, and even SAR-based algorithms cannot maintain a strong consistency with the observed values at the tidal gauge station. Nonetheless, it is evident that the MOPCA and SAMOSA+ algorithms proposed by us show more outstanding performance among the existing algorithms. To provide a more intuitive comparison of the differences between different algorithms, we processed the data and calculated the standard deviation and correlation coefficient with the observed values at the tidal gauge station, providing a quantitative analysis.

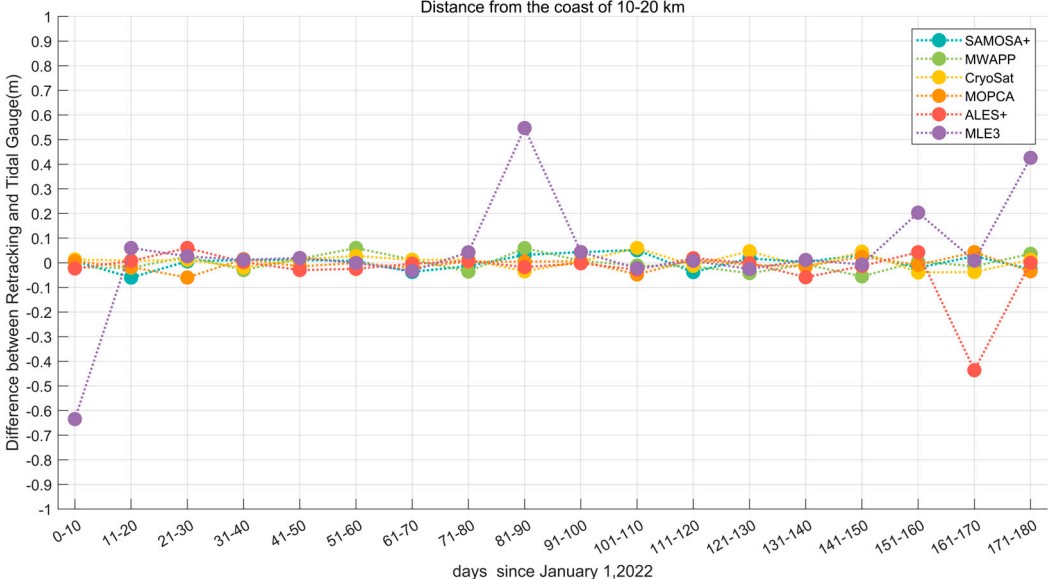

**Figure 19.** *Cont.*

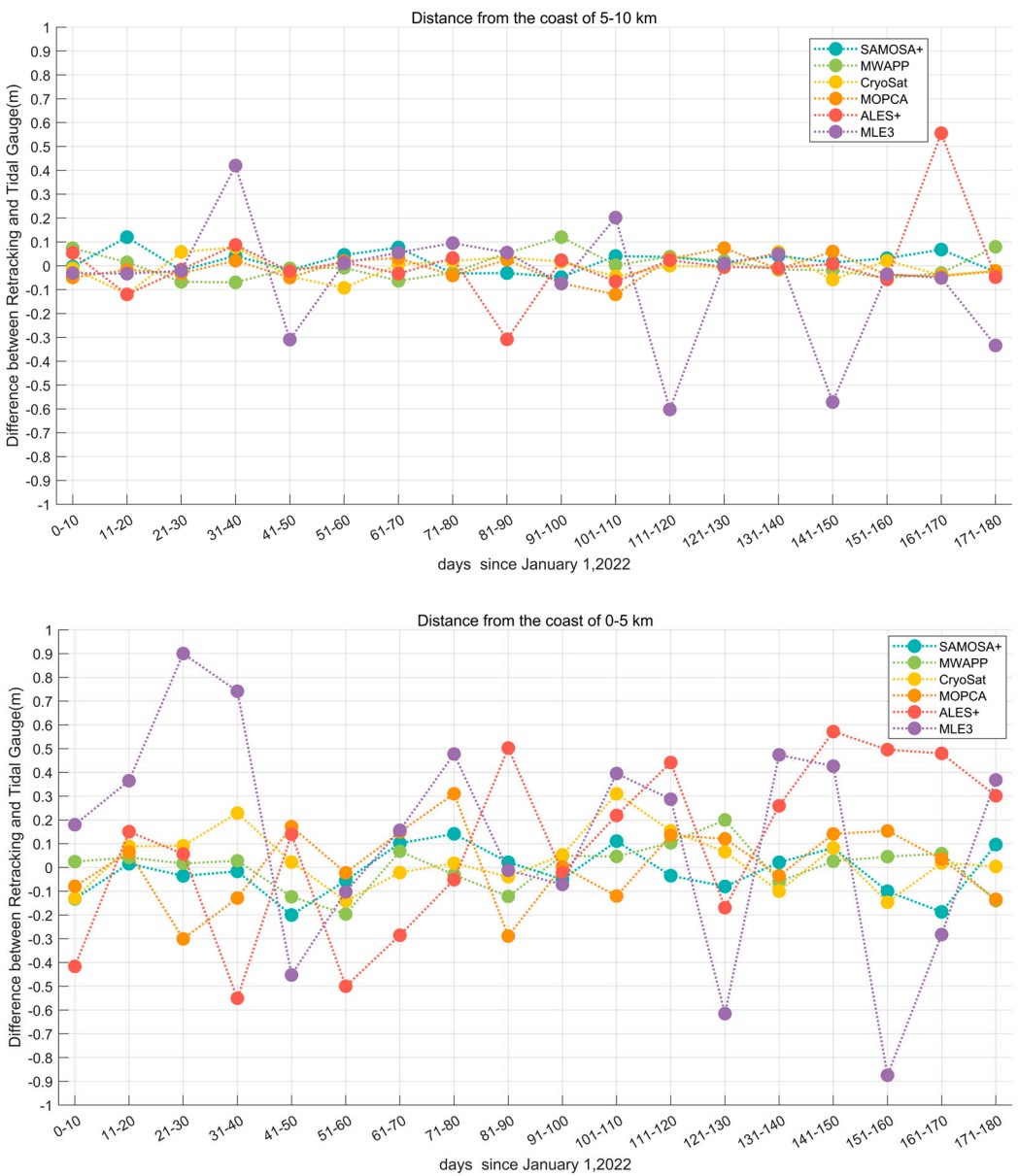

**Figure 19.** Time series (days) of the difference between various algorithms and the results from the nearest tidal gauge station are shown for distances from 10–20 km (**top**), 5–10 km (**middle**), and 0–5 km (**bottom**) to the coast.

The results presented in Figure 20 indicate that MOPCA and SAMOSA+ showed similar performance, with the highest correlation for distances within 4 km to the coast and lowest RMSE for distances within 20 km to the coast. The MWaPP and SAMOSA2 (Cryosat-2) algorithms showed similar performances, with slightly lower correlations and higher RMSE than the specialized coastal retrackers (MOPCA and SAMOSA+). The LRM retrackers (MLE3 and ALES+) showed the significantly lower correlation and highest RMSE for all distances within 10 km of the coast. These findings demonstrate that our proposed algorithm performs exceptionally well in the nearshore region, particularly within 10 km, and can provide more reliable nearshore sea level data. This can improve the applications of satellite altimetry technology in nearshore sea level changes and ocean gravity field modeling.

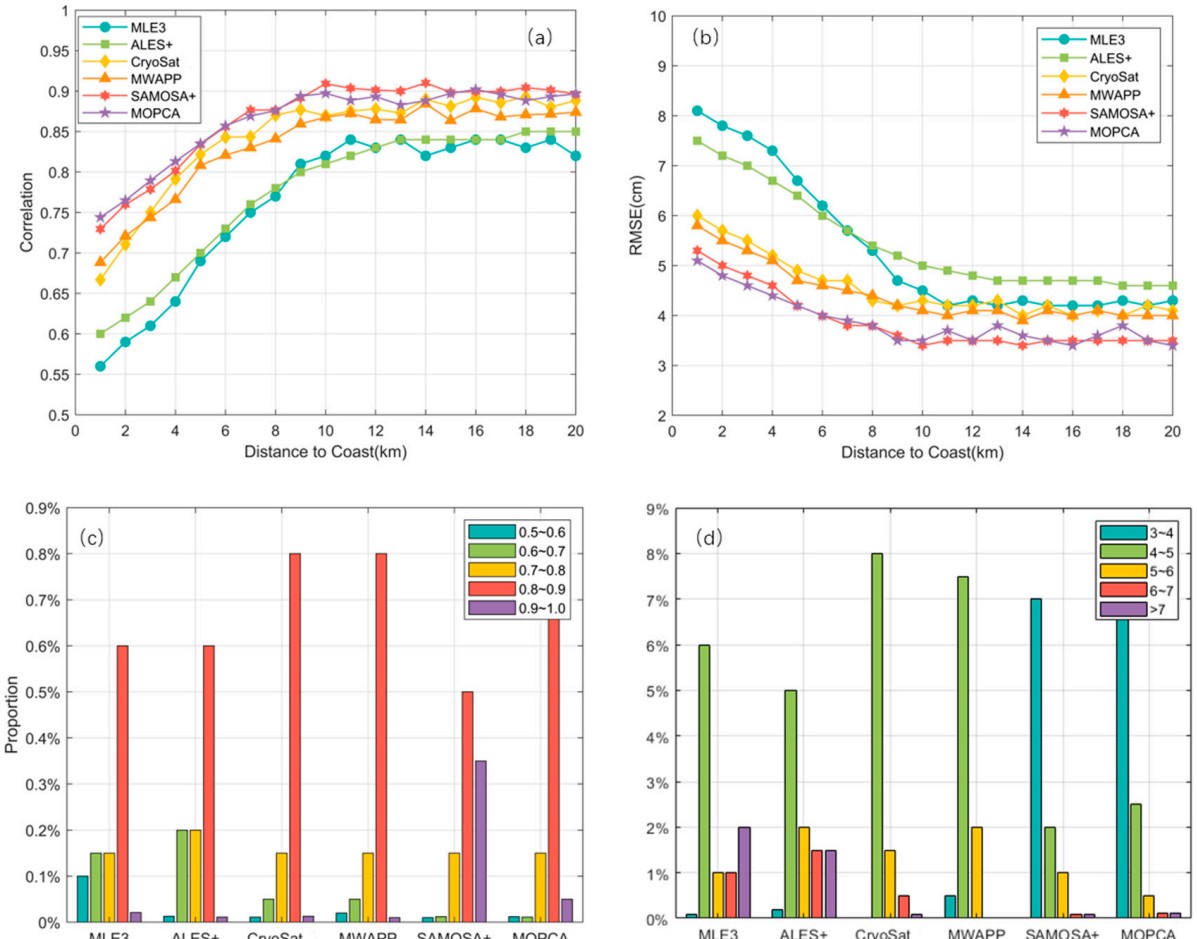

**Figure 20.** Plots illustrating the average correlation and RMSE time series between the different algorithms and the tidal gauge station sea level at various distances, as well as the proportion of correlation and RMSE within each interval. (**a**) Average correlation. (**b**) Average RMSE. (**c**) Proportion of correlation distribution within each interval. (**d**) Proportion of RMSE distribution within each interval.

## 6. Conclusions

To address the problems of decreased accuracy of satellite altimetry data in the nearshore region, reduced availability of data in coastal areas, and inability to process SAR data in real-time, we propose a coastal retracking strategy based on the parabolic cylinder model, named MOPCA. We have developed processing improvements from three aspects, namely, efficiency optimization, classification accuracy, and special processing for the coastal region. Finally, taking tidal gauge observations as reference, we compared the performance and accuracy of our proposed retracking strategy with other representative retracking algorithms. The results validated the performance and accuracy of the proposed retracking strategy.

The results show that compared with CryoSat, MWAPP, and SAMOSA+, the waveform retracking efficiency of MOPCA algorithm is greatly improved. For complex SAR waveform solutions, our algorithm uses a parabolic cylindrical model, whose simplicity leads to the processing speed comparable to traditional LRM waveform processing algorithms. The SAR waveforms can be tracked onboard with affordable computation resource. This will be a competitive algorithm for the generation of fast-delivery data products and can bring great benefits to applications such as storm surge monitoring.

Due to the simplicity of the parabolic cylindrical model's mathematical expression, the Bayesian algorithm can be employed instead of the commonly used least squares

algorithm for parameter estimation, and the precision of the height estimation can be improved significantly. The inclusion of Bayesian estimation is the core optimization of the MOPCA strategy.

Despite of the increasing calculation speed, MOPCA's percentage of waveforms successfully retracked is comparable to the representative SAR waveform retracking algorithms, and can even outperform the most popular SAMOSA+ algorithm (and others) in coastal areas within 5 km where the land contamination is particularly severe. These results sufficiently demonstrate the superiority of our scheme in the processing of nearshore altimetry data.

The main focus of our future work plan is to merge various algorithms to acquire more accurate altimetry measurements. There was a debate for decades whether one single waveform retracker algorithm is enough for coastal altimetry. Theoretically, there is no single algorithm suitable for all types of waveform; but in practice, merging multiple retrackers can usually cause spurious "jumps" in the SSH or SLA series, and the accuracy would thus be deteriorated because the different retrackers have different system biases (which are sensitive to parameters such as SWH). To achieve a consistent SLA series is a cumbersome problem, and the authors are now investigating the alignment of different algorithms (for example, the MOPCA and SAMOSA+ algorithms). The authors are also considering further improvements in the efficiency of MOPCA, to better meet the requirements for real-time processing. Considering that the main time-consuming operation of the MOPCA algorithm is the processing of the RNN part, developing a lightweight RNN algorithm will be another focus of our future work.

**Author Contributions:** Conceptualization, X.-Y.X. and J.Z.; methodology, J.Z. and X.-Y.X.; software, J.Z. and C.G.; formal analysis, J.Z. and X.-Y.X.; investigation, J.Z. and X.-Y.X.; data curation, J.Z. and C.G.; writing—original draft preparation, J.Z, X.-Y.X. and Y.X.; writing—review and editing, J.Z. and X.-Y.X.; project administration, X.-Y.X.; funding acquisition, X.-Y.X. All authors have read and agreed to the published version of the manuscript.

**Funding:** This research was funded by the National Natural Science Foundation of China (Grant No. 41876209).

**Data Availability Statement:** The Sentinel-6 radar altimeter L1B and L2 data are archived and maintained by EUMETSAT. The tide gauge data are downloaded from NMDIS (National Marine Data and information Service) in China.

**Acknowledgments:** The authors thank EUMETSAT for the Sentinel-6 altimetry data and NMDIS for the tide gauge data.

**Conflicts of Interest:** The authors declare no conflict of interest.

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
