# Peer review of "Coastal Waveform Retracking for Synthetic Aperture Altimeters Using a Multiple Optimization Parabolic Cylinder Algorithm"

_remotesensing, doi:10.3390/rs15194665_

Round 1
Reviewer 1 Report
This paper presents a valuable study and proposes a coastal retracking strategy (MOPCA) based on the parabolic cylinder model and Bayesian estimation. The paper is well conceived and the analysis are properly documented. However, I have some minor questions or suggestions to be checked by the authors.
(1) P2.L60-61, what are the exact PRFs for CrySat-2 and Sentinel-3?
(2) P3.Figure2&3, what are the data sources for these SAR and LRM coastal echo waveforms? Same mission (e.g. CS-2) or similar geolocation?
(3) P7.Figure 6, what dose author want readers to obtain from Figure 6? I suggest to add another reference curve for converntional parabolic cylinder model as reference to convince readers that the LUT-based optimization did not bring obvious changes.
(4)P7.Figure 7, are these two sample waveforms suficiently typical? The linewidth should be more thick for better distinguishment.
The same comment holds for P12 Figuire 14.
(5) P7. L167&170, typing errors. Figure 6 should be Figure 7.
(6)P9.Figure 9, what does group 1 to 5 represent for? the legend in subplot (b) should not be x1 to x5.
(7)P14.L351. HR should be high-resolution.
(8)P19.L480, the top and bottom were reversed.
Author Response
Thank you for your comments. These comments are all valuable and very helpful
for revising and improving our paper. We have fully addressed all the remarked points according to your suggestions in the report. The changes are marked in blue in the attachment. The main corrections in the paper and the responds to the reviewer’s comments are as follows .

Reviewer 2 Report
Review of Paper RS-2548841“Coastal Waveform Retracking for Synthetic Aperture Altimeters Using a Multiple Optimization Parabolic Cylinder Algorithm.”
By Jincheng Zheng, Xi-Yu Xu, Ying Xu and Chang Guo
Summary
This is an interesting paper with important new results investigating the performance of a new SAR re-tracker.
However, there are some major problems with the paper and a major revision is needed
Significant editing is required to the text, as there are some factual errors in the text.
The discussion and analysis in Section 5 is not well described, and requires major revision before it is acceptable for publication.
Many important references are missing. The authors should be careful to fully recognise other relevant work in this field.
General points
Section 1: Introduction
The authors do not report other recent developments in SAR altimeter re-trackers, and they should be careful to refer to relevant previous work. An expanded section should include suitable references and a discussion of other approaches.
Section 2: Parabolic cylinder model and its accelerated version algorithm.
Some specific comments below. A number of figures are not well described and/or incorrectly referenced.
Section 3: Coastal retracker strategy based on the parabolic cylinder model.
Again some specific comments below. A number of figures are not well described and/or incorrectly referenced.
Section 3.2 – Two-Step retracking. A very similar approach has previously been applied by Gao et al. (Remote Sens. 2019, 11, 718; doi:10.3390/rs11060718). The authors should refer to this previous work.
Section 4: Study Areas and Data
No specific comments
Section 5: Results
Section 5.2
The analysis uses terms such as performs “better” or “worse”. The authors should use more precise language. I.e. the retracker retracks a higher (or lower) percentage of waveforms; the retracker results give a lower (or higher) standard deviation of sea surface height; etc.
Section 5.2.3
The comparison against Tide Gauge data is not adequately described and is confusing.
This section should be rewritten.
Section 6: Conclusions
Some improvements to the discussion are recommended
Figures:
A number of figures have captions which do not adequately describe the figures, and some are not directly referred to in the text. A more careful proof-read before submission should have picked up these issues.
Please see attached file for detailed comments

Author Response

(The authors gave the same response as above.)

Round 2
Reviewer 2 Report
Summary
The authors have responded to the comments and have made important alterations to the text. I recommend publication after minor revision – see below
General points
There are some remaining small corrections required. Some are simple spelling / typing errors. I recommend a very careful final proof read in case I have not found all the mistakes.
Specific points
Line 175
The text refers to “right plot” of Figure 7, but there is only one panel.
Please correct the text.
Figure 9 – I still do not understand what is indicated by groups 1, 2, 3, 4, 5.
Please add some more detail to the description.
Section 5.2 line 411 – The text still refers to ““standard deviation of the retrackers altimetry results” – although the authors’ response says that it has been changed to “Standard deviation of the sea surface height of the retracked altimetry results”.
Please check again.
Figure 19 – Please explain the scale on the x-axis of the three panels of this figure.
Line 481, 482. Repeated use of correlation in this sentence. The first use is sufficient for the explanation.
Line 483 dash in the middle of the word SA-MOSA2. Please correct
Line 513 – misspelling of SMAOSA+ - please correct.
There are some remaining small corrections required. Some are simple spelling / typing errors. I recommend a very careful final proof read in case I have not found all the mistakes.
Author Response
Thank you again for your comments concerning the revised manuscript. These comments are all valuable and very helpful for revising and further improving our paper. We have fully addressed all the remarked points according to the editors’ and reviewers’ suggestions in the report.
